# Influence of Fly Ash Additive on the Properties of Concrete with Slag Cement

**DOI:** 10.3390/ma13153265

**Published:** 2020-07-23

**Authors:** Anna Szcześniak, Jacek Zychowicz, Adam Stolarski

**Affiliations:** Faculty of Civil Engineering and Geodesy, Military University of Technology, 2 gen. Sylwestra Kaliskiego Street, 00-908 Warsaw, Poland; jacekzychowicz@gmail.com (J.Z.); adam.stolarski@wat.edu.pl (A.S.)

**Keywords:** concrete, fly ash, slag cement, chloride ion migration, mechanical properties

## Abstract

This paper presents research on the impact of fly ash addition on selected physical and mechanical parameters of concrete made with slag cement. Experimental tests were carried out to measure the migration of chloride ions in concrete, the tightness of concrete exposed to water under pressure, and the compressive strength and tensile strength of concrete during splitting. Six series of concrete mixes made with CEM IIIA 42.5 and 32.5 cement were tested. The base concrete mix was modified by adding fly ash as a partial cement substitute in the amounts of 25% and 33%. A comparative analysis of the obtained results indicates a significant improvement in tightness, especially in concrete based on CEM IIIA 32.5 cement and resistance to chloride ion penetration for the concretes containing fly ash additive. In the concretes containing fly ash additive, a slower rate of initial strength increase and high strength over a long period of maturation are shown. In accordance with the presented research results, it is suggested that changes to the European standardization system be considered, to allow the use of fly ash additive in concrete made with CEM IIIA 42.5 or 32.5 cement classes. Such a solution is not currently acceptable in standards in some European Countries.

## 1. Introduction

Research on low-emission concrete, also called green concrete, is focused on waste recycling [1,2] and the reduction of Portland clinker in the concrete composition without affecting mechanical properties. Concrete additives, such as fly ash and blast furnace slag, are used for this purpose [3,4,5].

Fly ash (FA) is a by-product of the coal dust combustion process in heat plants and power plants, and after passing the certification process, it can be used as an additive to concrete. The use of FA and blast furnace slag as an additive to cement and concrete mixtures allows for waste recycling, reduces the cost of producing building materials, reduces the carbon footprint of concrete, and has a positive effect on selected physical properties of concrete.

Numerous experimental tests on concrete mixes have shown a significant improvement in the tightness of concrete after the application of FA additive. The positive effect of the addition of FA to cement and concrete mixes is closely related to the properties, quantity, and quality of the FA used [6]. Significant improvement in the concrete’s resistance to chloride penetration has been shown, provided that more than 20% FA is used in relation to the weight of cement [7,8,9,10]. Fly ash additive also causes an increase in the concrete’s resistance to high temperatures exposure, i.e., 200–400 °C [11]. Moreover, the results of experimental studies indicate increased the liquefaction of the concrete mix, easier pumpability, lower water absorption, and greater water tightness and resistance to the carbonation of concrete after using FA additive in comparison with ordinary concrete [12,13]. It should be noted that the increased liquefaction of the concrete mix after the addition of fly ash allows for its use in constructions that contain spatial concrete reinforcement. The addition of FA also reduces the calorific value of the concrete mix, which directly reduces the shrinkage effect and cement hydration temperature [14,15]. The results of experimental research by Rutkowska and Małuszyńska [16] indicate increased liquefaction of the concrete mix, as well as reduced water absorption and increased water tightness of concrete after using fly ash additive in comparison with ordinary concrete. The addition of FA can also have a positive effect on the mechanical properties of concrete. As reported by Golewski [17], an increase in the tensile strength of concrete containing 20% FA additive was shown. Simultaneously, the author noted that the addition of fly ash in the amount of 30% of the cement weight caused a sharp decrease in the tensile strength of concrete and an increase in crack propagation in the material. The positive effect of using 20% fly ash additive on the strength and crack resistance of concrete over a long maturation period, i.e., over 28 days, has been presented in several publications [18,19,20]. A similar relation was shown for concrete abrasion resistance in the paper of [21].

In [3], Golewski showed that a slowdown in the rate of concrete strength increases with the addition of fly ash when used as a partial replacement of Portland cement in amounts of 20% and 30% of the cement weight. For samples tested after 3, 7, and 28 days, the reduced compressive and tensile strength of concrete with the addition of fly ash was observed. Tests carried out on concrete samples after a long maturation period, i.e., over 90 days, showed an increase in the strength of concrete containing fly ash additive. The greatest slowdown in the rate of increase in strength was observed in concrete with 30% FA additive. The compressive strength of concrete with 20% fly ash additive after 365 days was 11–13% higher than that of concrete without FA additive, while the increase in tensile strength was 12%. In [3], the influence of the use of fly ash on fracture toughness was also examined. It was observed that the fracture toughness decreased as the amount of FA additive in the concrete increased. This effect was observed during a concrete maturation period of under 90 days. After 180 and 365 days, the fracture toughness was even higher for concrete containing 30% FA additive compared with reference concrete.

It is worth noting that fly ash can be obtained from various sources of waste recycling. Rutkowska et al. [22] studied the impact of fly ash from the combustion of municipal sewage sludge, as an additive to concrete on selected concrete properties. These studies indicated a significant increase in tightness and a slight decrease in the compressive strength of the concrete with fly ash additive over a long period of maturation.

To date, the conducted studies have focused on the properties of concrete with fly ash additive based on CEM I cement. The EN 206+A1:2016-12 standard [23] allows the addition of fly ash to concretes made with the use of CEM I and CEM IIA cements in amounts not exceeding 33% and 25% of the cement weight, respectively, with the ash coefficient k=0.4. However, the standard [23] does not provide for the addition of fly ash to concrete mixtures made using CEM III cement, containing blast furnace slag.

CEM III cement has the following properties:Increased stable strength and very good dynamics of strength increase over long periods of maturation;Low heat of hydration [24];High resistance to the action of corrosion factors, especially sulfates [24];Lower water and salt absorption [24];Resistance to chloride penetration and reinforcement corrosion [24];Low capillary porosity [25];Good workability of the mix;Stability of volume;Resistance to the destructive action of alkali from cement in concrete with reactive aggregate [24];Light color, allowing for its use in the production of architectural concrete.

Van Noort [26] indicated that the quantity and origin of slag have a significant impact on the resistance to corrosive factors of concrete made using slag cement. In addition, concrete made using slag cement is exposed to increased carbonation. This effect mainly concerns CEM IIIB cement. Experimental studies [24] have indicated that, in the case of concretes made with CEM IIIA cement, where the slag content does not exceed 57%, the phenomenon of carbonation is comparable to that in concretes made with CEM I cement.

Because there is a significant amount of blast furnace slag in proportions from 35% to 65%, CEM III cement can be classified as an ecological cement binder, with a low carbon footprint and high recyclability properties. Slag is a by-product in the metallurgy industry resulting from the smelting of iron ore in a blast furnace. Because of its hydraulic properties, blast furnace slag is a valuable cement component. The use of blast furnace slag in cement allows the amount of Portland clinker to be reduced, which is favorable for ecological reasons. The production of Portland clinker is a highly energy-intensive process that has a negative influence on the environment [27].

Table 1 summarizes the standard guidelines related to the permissibility of using fly ash additive with CEM III cement, in various European countries.

The main areas of application of concrete with CEM III binder are foundation slabs, massive foundation footings, diaphragm walls, and structural elements of underground tanks. This is due to the need to provide low concrete shrinkage, resistance to sulfate aggression, or a low heat of hydration. Foundation slabs are designed in exposure class XD3 because of their susceptibility to corrosion due to chloride. According to the standard EN 206+A1:2016-12 [23], for exposure class XD3, the minimum concrete strength class required is C35/45, the maximum water-cement ratio is *w/c* = 0.45, and the minimum content of CEM I or CEM IIA cement is 320 kg/m^3^.

Properties resulting from the use of fly ash additive to concrete include:Improved workability and stability of the concrete mix;Reduced effects of water release from concrete—less “bleeding”;Increased tightness.

These characteristics are extremely important when making foundation slabs.

In [10], Bijen draws attention to the significant benefits of using concrete with the addition of fly ash and blast furnace slag in reinforced concrete structures. Because of the advantages, such as the significant reduction in the rate of chloride ion penetration into concrete and the increase in the critical chloride concentration associated with chloride-induced corrosion concrete, structures can have a long corrosion-free service life under the most aggressive environmental conditions, in which the concrete is the only protection for the reinforcement.

The purpose of this research is to demonstrate how the use of fly ash as an additive constituting more than 25% by weight of CEM IIIA N LH HSR NA cement with strength classes 32.5 and 42.5 affects the physical and mechanical properties of concrete; in particular, its tightness and the migration of chloride ions.

It should be noted that currently, in some countries, such as Poland, Denmark, and UK, the use of fly ash for concrete based on CEM III cement is not allowed by the standard [23] and its national equivalents. Despite this objection, the justification for undertaking research on this subject is the fact that, in other European countries, such as Germany, the Czech Republic, and Slovakia, and in countries with similar climatic conditions, such as Belgium and the Netherlands, the use of fly ash additives for concrete mixes made of CEM III cement is allowed. In the German standard DIN 1045-2:2014-08 [28], the limits regarding the composition and properties of concrete in the XD3 exposure class in relation to the concrete strength class, the maximum value of the factor w/(c+k×fa)=0.45, and the minimum cement content are the same as those in the Polish and British standards [23]. In the standard [28], the minimum cement content for concrete without additives is 320 kg/m^3^ for the XD3 exposure class. However, for concrete with additives, the minimum cement content was reduced to 270 kg/m^3^. Simultaneously, the standard [28] allows the use of an additive in the form of fly ash for concrete made of CEM IIIA and CEM IIIB cement (with a maximum slag content of 70%), with a value of k=0.4.

The results of the tests carried out in this work indicate a positive influence of fly ash additive on the properties of concrete made by using CEM IIIA LH HSR NA cement of strength classes 32.5N and 42.5N. The results of experimental tests that measured strength properties and resistance to water and chloride ion penetration for concrete made of CEM IIIA LH HSR NA cement with strength classes 32.5N and 42.5N, with and without fly ash additive, were analyzed. The experimental results show that it is possible to obtain concrete based on slag cement with fly ash additive with very good tightness and resistance to chloride corrosion without impairing concrete’s strength properties. The tests were carried out for the addition of fly ash, constituting 25% and 33% of cement weight. The justification for the use of such an amount of fly ash additive is the guidelines of standard EN 206+A1:2016-12 [23], for the maximum use of fly ash additive. However, these guidelines only refer to the possibility of using cement CEM I.

## 2. Materials and Methods

The samples for testing were prepared for three series of concrete mixes, using CEM IIIA 42.5N cement and three series of concrete mixes made using CEM IIIA 32.5N cement. The series of concrete mixes differed in the amount of fly ash additive substituted for slag cement. 

For testing, the following materials were used:Slag cement CEM IIIA 42.5 and CEM IIIA 32.5;Fly ash;Fine aggregate—pit sand with a maximum size of 2.0 mm;Coarse aggregate—natural gravel with a minimum and maximum size of 2.0 and 8.0 mm;Lignosulfonate- and naphthalene-based admixture;Polycarboxylate-based admixture;Pure laboratory pipeline water.

### 2.1. Cements

CEM IIIA 42.5N LH HSR NA and CEM IIIA 32.5N LH HSR N slag cements (Górażdże Cement S.A., Chorula, Poland) were used for testing the concrete series, denoted by CM1 and CM2, respectively. The physical parameters, mechanical parameters, and chemical properties of these cements, made available by the producer, are presented in Table 2.

### 2.2. Fly Ash

As an additive to concrete, fly ash originating from hard coal combustion in the Kozienice CHP plant located in central Poland (Kozienice) was used. The used FA has pozzolanic properties, consists mainly of spherical grains, and is certified in the 1+ system, meeting the requirements of the EN 450-1 standard [29].

FA was characterized by the following parameters:Fineness: category N, below 12%;Loss on ignition: category A, below 5%.

The detailed results of the chemical and phase composition (Figure 1) of fly ash from the Kozienice CHP plant are presented in [30]. The phase composition of FA from Kozienice, determined using the X-ray diffraction method, indicates that the content includes the following chemical compounds:Mullite A_3_S_2_;Quartz SiO_2_;Hematite Fe_2_O_3_;Magnetite Fe_3_O_4_.

The compounds contained in the phase composition are listed in order of decreasing percentage share of FA weight.

### 2.3. Concrete Mixes

The tests were carried out for six series of concrete mixtures made using CEM IIIA 42.5N cement (designations CM1-0, CM1-25, CM1-33), and using CEM IIIA 32.5N cement (designations CM2-0, CM2-25, CM2-33). The reference concrete mixes CM1-0 and CM2-0 meet the requirements of exposure class XD3 and were designed assuming the value of *w/c* = 0.45 without the addition of fly ash. Next, the composition of reference concrete mixes was modified by adding fly ash, with the simultaneous reduction of cement and sand aggregate and the assumption of unchanged w/(c+k×fa)=0.45. Then, the reference concrete mixes were modified by using fly ash as a partial cement replacement. In the concrete mixes CM1-25 and CM2-25, the fly ash content was 25% of the cement weight, while the maximum amount of fly ash, i.e., 33% by weight of cement, was used in the CM1-33 and CM2-33 concrete mixes. This quantity is allowed in the standard [23], only for CEM I cement. The composition of concrete mixes is presented in Table 3.

## 3. Test Methods

### 3.1. Concrete Mix Consistency Testing

The consistency of the concrete mix was tested using the falling cone method, in accordance with EN 12350-2 [31].

### 3.2. Research on Migration of Chloride Ions in Concrete

The chloride ion migration coefficient in hardened concrete was determined according to the method presented in the NT BUILD 492 standard [32]. The tests were carried out for concrete samples of the CM1-0, CM1-25, CM1-33 series. Three disks with a diameter of 100 mm and a height of 50 mm were cut from each of the samples and subjected to subsequent tests. To determine the resistance of hardened concrete to the penetration of aggressive chloride ions, the chloride ion migration factor *D_nssm_* was used, determined in accordance with [32]. The chloride ion migration coefficient was determined from the transient chloride ion flow caused by an external electric field of a given voltage. Before testing, the side edges of the disks were protected with epoxy resin and saturated with a Ca(OH)_2_ solution. The value of the *D_nssm_* coefficient was determined on the basis of the voltage applied, the temperature of the anolyte measured at the beginning and end of the test, as well as the depth at which chloride ions penetrated, which was measured on an axially split sample. The study was conducted within 48 h.

The value of the chloride ion migration coefficient in the transient state was calculated according to the NT BUILD 492 standard [32], using the following expression:(1)$$D_{nssm}=\frac{0.0239\left(273+T\right)L}{\left(U-2\right)t}\left(x_{m}-0.0238\sqrt{\frac{\left(273+T\right)Lx_{m}}{U-2}}\right)\left(\times 10^{-12}\frac{m^{2}}{s}\right)$$
where

*U*—value of voltage used (V);

*T*—average value of the initial and final temperature of the anode liquid (°C);

*L* = 50 mm—sample height;

$x_{m}$—average penetration depth value (mm);

*t*—duration of the test (h).

### 3.3. Concrete Waterproof Test

Concrete water tightness testing for samples of all series was carried out after 120 days of maturation. The tests were performed on cubic samples with a side dimension of 150 mm, whose side surfaces were protected with waterproofing resin. The samples were exposed to pressurized water. For this purpose, an apparatus for testing the water permeability of concrete 55-C0246/6 (Controls, Liscate, Italy) was used. The device consists of a steel frame with clamps with a hydraulic system, valves, a water pressure indicator, and transparent measuring burettes. The apparatus is adapted for simultaneous testing of up to six cubic samples, and enables water supply at a maximum pressure of 10 bar. The water pressure was increased every 24 h by 0.2 MPa, in the range of 7.4–8.0 MPa. The final pressure, corresponding to a water resistance of 80 m, was maintained for 24 h. The samples were then split, and the depth of water penetration into the concrete was measured using an electronic caliper. From the results of measurements of the depth of water penetration into concrete, the average depth of water penetration into concrete was calculated for the tested samples:(2)$$w_{m}=\frac{1}{n}\sum_{i=1}^{n}w_{i}$$
where

*w_m_*—average depth of water penetration into concrete (mm);

*w_i_*—maximum depth of water penetration into concrete in the *i*th sample;

*n*—number of samples tested.

### 3.4. Testing of Compressive Strength of Concrete

Cubic samples with side dimensions equal to 100 mm were tested to measure the concrete compressive strength. The tests were carried out for concrete recipes CM1-0, CM1-25, CM-33 and CM2-0, CM2-25 and CM2-33 in subsequent periods of concrete maturation from 1 to 234 days. Samples were stored in accordance with the standard [33]. The tests were carried out in accordance with the standard [34], using a MEGA 6-3000-150 (Form+Test, Riedlingen, Germany) hydraulic press.

### 3.5. Testing of Tensile Strength of Concrete During Splitting

Concrete tensile strength at splitting was tested in accordance with the standard [35]. Six cubic samples with side dimensions equal to 150 mm were tested in each series. The tensile strength of the concrete during splitting was determined from the following relationship:(3)$$f_{ct,spl}=\frac{2F}{\pi d^{2}}$$

*f_ct,spl_*—tensile strength of concrete during splitting (MPa);

*F*—splitting force (N);

*d*—dimension of the sample side (mm).

## 4. Test Results

### 4.1. Concrete Mix Consistency

The results of consistency tests presented in Table 4 indicate that the addition of FA caused an increase in the flowability of the concrete mix, in the case of concrete based on C32.5 cement. However, the consistency class for all series of C42.5-based concrete mixes remained unchanged.

### 4.2. Concrete Resistance to Chloride Ion Penetration

A comparative analysis of the obtained results of the chloride ion migration test for the three analyzed recipes indicates a significant increase in resistance to the penetration of aggressive chlorides into concrete in which FA additive was used. The measurement parameters used during the testing are shown in Table 5.

The category of resistance to the penetration of chloride ions with the designation “good” was obtained, both in the case of concrete without the addition of FA, i.e., CM1-0, and in the case of concrete CM1-25, in which FA was used in the amount of 25% of the cement weight. However, comparing the average values of the chloride ion migration coefficient, one can notice a clear reduction in its value, in the case of concretes containing FA additive. The average value of the coefficient obtained for CM1-25 concrete is 26% lower than the value of the coefficient obtained for CM1-0 concrete, while for CM1-33 concrete containing the largest amount of ash (33% of the cement weight), the coefficient value is as much as 43% lower. In addition, CM1-33 concrete was classified in the highest category of resistance to chloride ion penetration, defined as “very good”. The reduction in the chloride ion migration coefficient for concrete samples containing FA additive is directly related to the average depth of chloride ion penetration. In CM1-0 concrete, the depth of ion penetration is almost double that of CM1-33 concrete.

Figure 2 and Table 6 summarize the results obtained when determining the chloride ion migration rate in hardened concrete, for the three analyzed recipes, CM1-0, CM1-25, and CM1-33.

One of the results obtained for the CM1-33.3 series was rejected because of microcrack induction.

### 4.3. Water Permeability of Concrete

During the tests, no water permeation through the concrete sample was found in any case. The results in Table 7 refer to the maximum water penetration depths that were recorded in the sample. It should be emphasized that the maximum water penetration depth was recorded locally. The comparative analysis of the results presented in Figure 3 and Table 7 indicates a reduction in the depth of water penetration into the concrete after applying FA. As the amount of FA additive increases, the average water penetration depth into the concrete decreases. The use of FA increased the tightness of concrete based on CIIIA 32.5 cement to a greater extent, compared with concrete based on CIIIA 42.5 cement.

The mean value of the water penetration depth *w_m_* determined for concrete with the addition of less ash (CM1-25) is 10% smaller than the value determined for concrete without the addition of ash (CM1-0), and with the maximum ash addition (CM1-33), it is smaller by as much as 19%. In turn, in concretes made on the basis of CEM IIIA 32.5 cement, the addition of FA causes a decrease in the average value *w_m_* by 68% for CM2-25 concrete and by as much as 87% for CM2-33 concrete compared with the average value *w_m_* obtained for concrete without FA addition, i.e., CM2-0.

### 4.4. Compressive Strength of Concrete

The results of the compressive strength tests are summarized in Figure 4. The results show a clear delay of increase in the compressive strength of concrete after applying FA additive. This phenomenon is most intensified with a maturation time of less than 28 days for CM1 concrete and less than 56 days for CM2 concrete. It should be noted that the early strength of CM2-0 concrete on the 4th day of maturation is higher than the strength of CM1-0 concrete. After 56 days of concrete maturation, the difference in the concrete strength of CM1-0 and CM1-33 does not exceed 5%. However, for concrete CM2-0 and CM2-33, this difference is 14%. For the long maturation period, the difference in the strengths of the CM1-0, CM1-25, and CM1-33 concretes remain at the same level, and the reduction in strength increases with the amount of FA added.

Among CM2 concretes with long-term maturation, i.e., more than 90 days, the strength of CM2-33 concrete is observed to increase in relation to CM2-25 concrete by 4%, as shown in Figure 4.

According to the results of this research, it can be stated that FA, as an addition to concrete based on slag cement, causes a decrease in the strength of concrete, after both the early and the long-term maturation of concrete. However, the use of FA additive in amounts above 30% in concrete based on CEM IIIA 32.5 cement does not rapidly cause a reduction in concrete compressive strength, as was observed in the case of concrete based on Portland cement (see, e.g., [6]).

The impact of FA addition on the development of compressive strength after long-term concrete maturation was also analyzed. The results of this analysis are shown in Figure 5, which presents changes in the mean values of the compressive strength of concrete in relation to the strength obtained on the 28th day of maturation, which was determined to be 100%. Concretes of the CM1 series are characterized by a similar increase in strength, which, after 194 days of maturation, does not exceed 23% compared with the strength at 28 days. In the CM2 series of concretes, the FA addition causes a significant increase in the concrete’s compressive strength with long-term maturation. The largest increase in strength is recorded for CM2-25 concrete, which, after 234 days of maturation, is 49% greater than the strength of the 28-day concrete.

### 4.5. Tensile Strength of Concrete at Splitting after Long Maturation Times

Figure 6 and Table 8 show the results of concrete tensile testing at splitting. The results obtained for the concretes of the CM1 series are similar. The strength of CM1-0 concrete is almost equal to the strength obtained for CM1-25 concrete and 6.5% greater than the strength obtained for CM1-33 concrete containing FA additive. Among CM2 concretes, the highest value of concrete tensile strength at splitting is obtained for CM2-25 samples, containing 25% FA addition. This strength is 44.6% higher than the strength recorded for CM2-0 concrete, which does not contain FA additive, and 10.3% higher than the strength obtained for CM2-33 concrete, containing 33% FA additive.

Considering the concrete maturation time, which was 120 days, it can be concluded that the influence of the FA addition on the tensile strength of concrete made from slag cement is consistent with the results obtained for concrete based on Portland cement (e.g., in [17]).

## 5. Discussion

The conducted tests indicate the usefulness of using type II additive in the form of FA for concrete containing blast furnace slag cement. The presented test results indicate that the addition of FA to concrete mixtures based on slag cements leads to an increase in concrete tightness and resistance to chloride aggression. Both these parameters are particularly important when making such structural elements as foundation slabs. The use of tight concrete resistant to the penetration of adverse and aggressive external factors is extremely important, not only for concrete durability but also for protecting reinforced concrete structures against corrosion. Limitation in the penetration of chloride ions is particularly important for protecting reinforcement steel against corrosion. Corrosion caused by chlorides is extremely dangerous because of its initially asymptomatic course. Consequently, it causes the formation of permanent damage to the reinforcement, which can lead to the sudden destruction of the structure. The phenomenon of corrosion caused by chlorides is additionally enhanced by moisture. For the analyzed concrete recipes, after using FA additive, a significant increase in the resistance of concrete to the penetration of chloride ions and water was observed. An increase in these parameters was observed with an increase in the amount of FA used. The results of the conducted tests indicate a significant reduction in the penetration of concrete by water after the application of FA additive for concretes using CEM IIIA cements, of both classes 42.5 and 32.5. The relationship between the depth of penetration of water and the depth of penetration of chloride ions into concrete was determined. Significantly limiting the depth of water penetration into concrete based on CEM IIIA 32.5 and 42.5 slag cements by using FA additive will significantly reduce the penetration depth of chloride ions, which is particularly important in the case of structures that are designed for exposure class XD3 and exposed to substantial chloride aggression.

After a short maturation time, the FA addition significantly reduced the concrete’s compressive strength in both CM1 and CM2 concretes. With the elongation of maturation time, an increase in the strength of concrete containing FA additive was observed. Moreover, in the second phase of maturation, from day 56, the CM2-33 concrete based on CEM IIIA 32.5 cement and containing a maximum of 33% FA additive was found to have higher strength than the CM2-25 concrete containing 25% FA additive. Comparing the results of the tests, it can be seen that concrete based on CEM III 32.5 cement with the addition of FA has a much greater resistance to water penetration and a greater reduction in strength in the initial maturation phase than concrete based on CEM III 42.5 cement with the same percentage of FA additive. In concrete with CEM III 32.5 cement, neither compressive strength nor tensile strength at splitting decreases proportionally to the increase in the FA additive content, as is observed with concrete with CEM IIIA 42.5 cement. The test results also indicate the usefulness of using FA as an additive to concrete based on CEM IIIA 32.5 cement, since there is a significant improvement in concrete tightness. Nevertheless, it should be remembered that the strength of concrete is reduced during the initial time of concrete maturation, which must be considered when designing it. In light of the test results obtained, changing the national standardization systems to allow the addition of fly ash to concretes based on CEM IIIA 42.5 and 32.5 slag cements is an issue worth considering.

## 6. Conclusions

The following conclusions were made from the results obtained during the research.

The addition of fly ash increases the tightness and resistance to the penetration of chloride ions of concrete made of slag cement.Fly ash reduces the depth of penetration of water under pressure into concrete based on CEM IIIA 32.5 slag cement, to a greater extent than that based on CEM IIIA 42.5 cement. The addition of FA reduced the depth of water penetration under pressure by 19% for CM1-33 concrete, and by as much as 87% for CM2-33 concrete, compared with reference concretes that did not contain fly ash.The relationship between the resistance of concrete to water penetration under pressure and resistance to chloride ion penetration is demonstrated. A comparison of the test results obtained for concretes CM1-0, CM1-25, and CM1-33 indicates that the effect of FA addition on reducing the mean value of the water penetration depth under pressure into concrete *w_m_* is proportional to the decrease in the average depth of the penetration of chloride ions into concrete *x_m,tot_*. The proportionality coefficient for this reduction is in the range $\frac{w_{m}}{x_{m,tot}}$ = (2.6,2.7).FA addition to concrete based on slag cement reduces compressive strength during the initial and long-term maturation times. In the case of concrete based on CEM IIIA 42.5 cement, the reduction in compressive strength after the application of FA additive in amounts of 25% and 33% by the weight of cement is similar and does not exceed 5%. This applies to both the standard maturation period, i.e., after 28 days, as well as the extended maturation period, i.e., after 190 days. Concrete based on CEM IIIA 32.5 slag cement after using FA additive is characterized by a significant reduction in compressive strength, which after 28 days of maturation, is 23% (series CM2-25). After the long maturation period of over 120 days, the reduction in compressive strength of concretes containing FA additive does not exceed 10%.The use of FA additive for concretes based on slag cement in amounts of up to 33% by weight of cement is beneficial. The conducted research shows that the FA addition of 33% compared with the addition of 25% by the weight of cement in concrete based on CEM IIIA 42.5 cement does not significantly affect compressive strength and, in the case of concrete based on CEM IIIA 32.5 cement, even causes an increase in this strength. These observations refer to both standard and long maturation periods. The increased addition of fly ash significantly increases the tightness of concrete and its resistance to the penetration of chloride ions.Consideration should be given to allowing the use of the *k* factor at XD1-3 concrete exposure for CEM IIIA cement classes as well.

## Figures and Tables

**Figure 1 materials-13-03265-f001:**
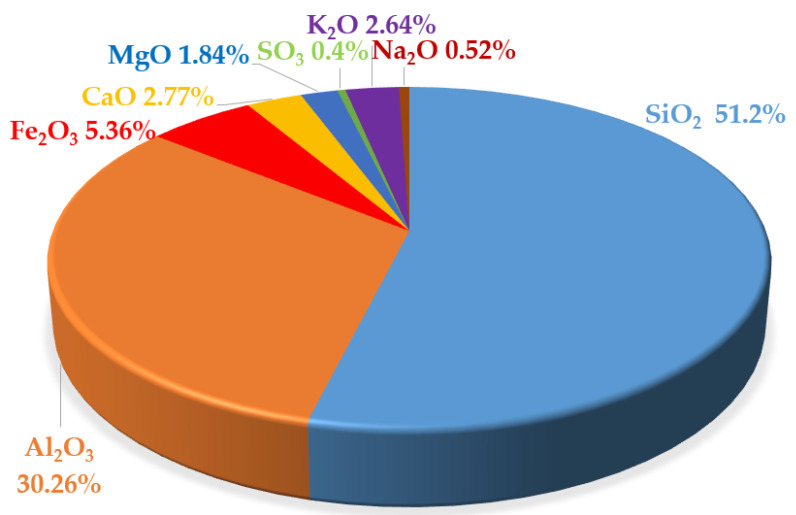
Chemical composition of Kozienice Fly Ash (on the basis of the data included in [30]).

**Figure 2 materials-13-03265-f002:**
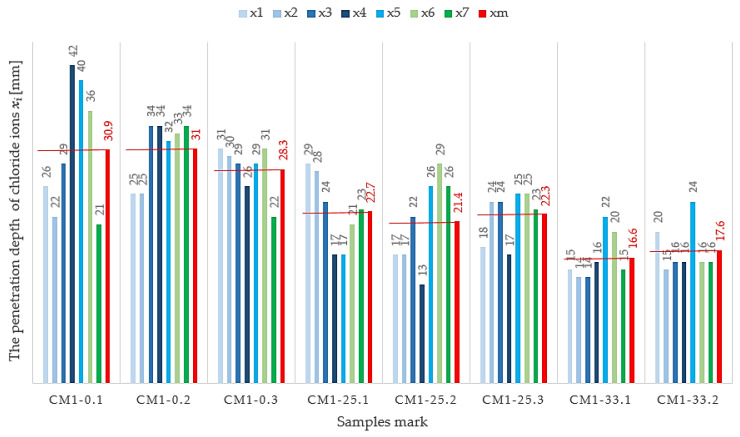
The penetration depth of chloride ions in the tested concrete samples.

**Figure 3 materials-13-03265-f003:**
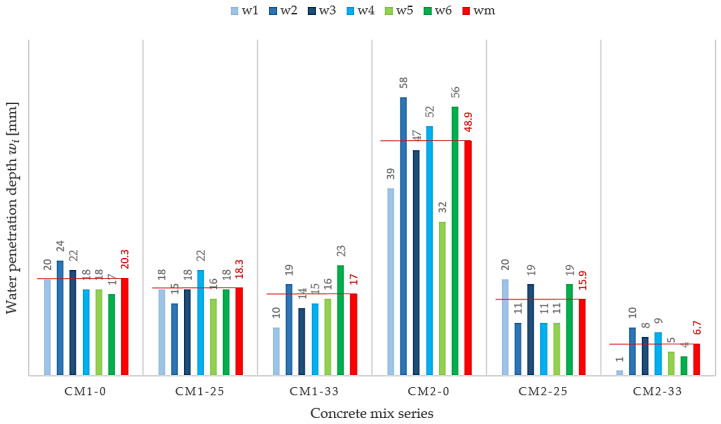
The depth of penetration of water under pressure into concrete for the tested mix series.

**Figure 4 materials-13-03265-f004:**
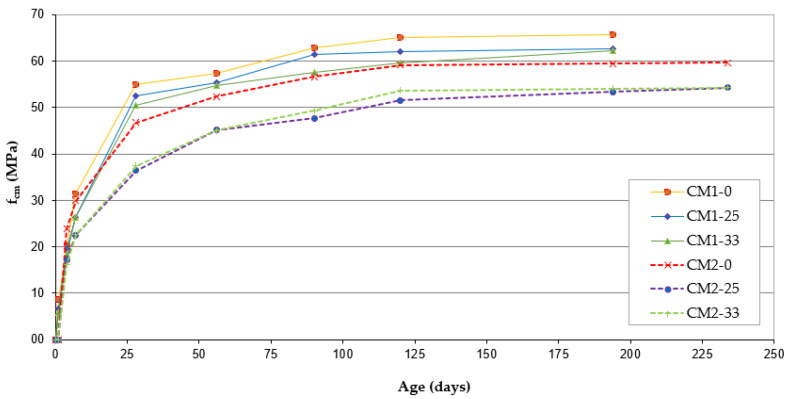
Compressive strength means values of concretes as a function of maturation time.

**Figure 5 materials-13-03265-f005:**
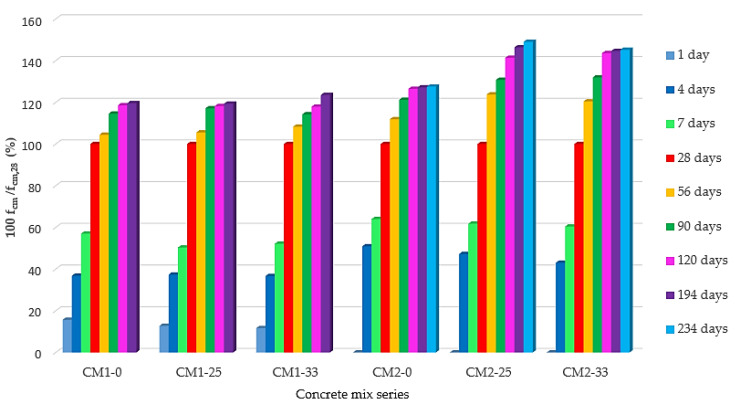
Relative changes in mean values of the compressive strength of concretes in relation to compressive strength in 28 days.

**Figure 6 materials-13-03265-f006:**
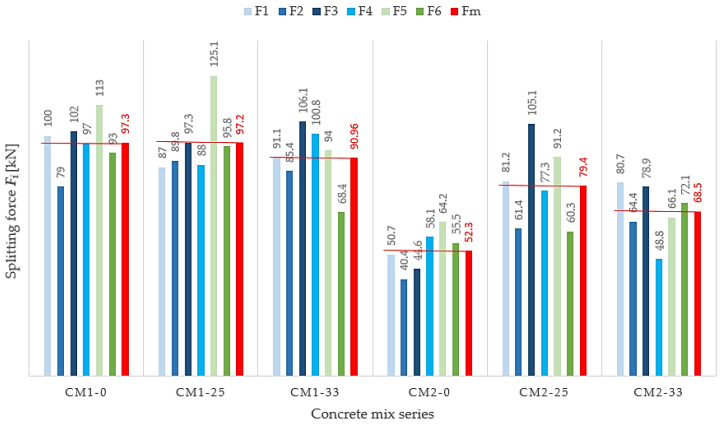
Splitting force of concrete for the tested mix series.

**Table 1 materials-13-03265-t001:** European countries where the use of CEM III cement in concretes with fly ash additive is allowed.

Country	Type of Cement	*k*^1^ Factor Value
Belgium	CEM IIIA	0.2 (FA/cement ≤ 0.25)
Czech Republic	CEM IIIA	0.2
Germany	CEM IIIA, CEM IIIB (slag content below 70%)	0.4 0.7 (for underwater concrete and drilled piles)
Italy	CEM IIIA	0.2
Netherlands	CEM IIIA, CEM IIIB	0.2
Slovakia	CEM IIIA	0.2
Finland	CEM I, CEM II/A-S, CEM II/A-D, CEM II/A-V, CEMII/A-LL, CEM II/A-M, CEM II/B-S, CEM II/B-V,CEMII/B-M, CEM III/A, CEM III/B	0.4

^1^ According to the standard EN 206+A1:2016-12 [23], the k-value concept permits type II additions to be taken into account by replacing the term ‘water/cement ratio’ with ‘water/(cement + *k* addition) ratio’.

**Table 2 materials-13-03265-t002:** Physical, mechanical, and selected chemical parameters of cements.

Component	Content	
	CEM IIIA 42.5N LH HSR NA	CEM IIIA 32.5N LH HSR NA
Specific surface—Blaine (cm^2^/g)	4196	4193
Start of setting time (min)	221	272
Heat of hydration (J/g)	227	171
Compressive strength after 2 days (MPa)	14.1	-
Compressive strength after 7 days (MPa)	-	20.9
Compressive strength after 28 days (MPa)	53.7	50.0
Sulfate SO_3_ (%)	2.19	1.48
Chloride Cl^−^ (%)	0.06	0.037
Alkali as Na_2_O_eq_ (%)	0.73	0.66
Loss on ignition (%)	1.11	0.81

**Table 3 materials-13-03265-t003:** Concrete mix designs.

Concrete Mix Series	Weight of Concrete Ingredients (kg/m^3^)							
	CEM IIIA 42.5	CEM IIIA 32.5	Fly Ash	Sand	Gravel	Water	Admixture 1 ^1^	Admixture 2 ^2^
CM1-0	320	0	0	740	1150	144	3.20	0.64
CM1-25	291	0	73	675	1150	144	3.28	1.09
CM1-33	283	0	93	655	1150	144	3.20	1.32
CM2-0	0	320	0	740	1150	144	3.20	0.64
CM2-25	0	291	73	675	1150	144	3.28	1.09
CM2-33	0	283	93	655	1150	144	3.20	1.32

^1^ Lignosulfonate- and naphthalene-based admixture; ^2^ polycarboxylate-based admixture.

**Table 4 materials-13-03265-t004:** Summary of concrete mix consistency testing results.

Concrete Mix Series	Cone Fall (mm)	Class of Consistency
CM1-0	150	S3
CM1-25	150	S3
CM1-33	140	S3
CM2-0	150	S3
CM2-25	170	S4
CM2-33	200	S4

**Table 5 materials-13-03265-t005:** Measurement parameters for the testing of the chloride ion migration coefficient.

Samples Mark	Initial Current (mA)	Corrected Voltage (V)	Corrected Current (mA)	Final Current (mA)	Anolyte Initial Temperature (°C)	Anolyte end Temperature (°C)
CM1-0.1	8.1	60	16.6	26.2	23.7	19.9
CM1-0.2	9.0		18.7	35.4	24.5	20.1
CM1-0.3	7.9		16.4	23.2	23.4	20.1
CM1-25.1	8.9	60	18.2	40.0	22.9	25.4
CM1-25.2	7.5		15.2	20.1		24.6
CM1-25.3	7.6		15.4	20.1		24.3
CM1-33.1	6.1	60	12.5	16.6	23.0	24.8
CM1-33.2	6.7		13.7	22.7		25.0

**Table 6 materials-13-03265-t006:** Test results for determination of the chloride ion migration coefficient.

Samples Mark	$x_{m}$	$x_{m,tot}$	$D_{nssm}\times 10^{-12}\left(\frac{m^{2}}{s}\right)$	$D_{nssm,m}\times 10^{-12}\left(\frac{m^{2}}{s}\right)$	Standard Deviation	Category of Chloride Ion Penetration Resistance
CM1-0.1	30.9	30.1	3.6	3.5	0.2	Good
CM1-0.2	31.0		3.6			
CM1-0.3	28.3		3.3			
CM1-25.1	22.7	22.1	2.7	2.6	0.1	Good
CM1-25.2	21.4		2.5			
CM1-25.3	22.3		2.6			
CM1-33.1	16.6	17.1	1.9	2.0	0.1	Very good
CM1-33.2	17.6		2.0			

**Table 7 materials-13-03265-t007:** Summary of the results of testing the depth of penetration of water under pressure into concrete.

Concrete Mix Series	Water Penetration Depth *w_i_* (mm)			
	*w_i,min_*	*w_i,max_*	*w_m_*	Standard Deviation
CM1-0	17	24	20.3	2.8
CM1-25	15	22	18.3	2.7
CM1-33	10	23	17.0	4.9
CM2-0	32	58	48.9	10.1
CM2-25	11	20	15.9	4.6
CM2-33	1	10	6.7	3.5

**Table 8 materials-13-03265-t008:** Tensile strength of concrete at splitting.

Concrete Mix Series	Splitting Force Mean Value *F_m_*	Splitting Force Standard Deviation	*f_ct,spl_* (MPa)
CM1-0	97.3	11.2	2.75
CM1-25	97.2	14.3	2.75
CM1-33	90.96	13.2	2.57
CM2-0	52.3	8.8	1.48
CM2-25	79.4	17.3	2.14
CM2-33	68.5	11.7	1.94

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
