# Peer review of "Influence of Fly Ash Additive on the Properties of Concrete with Slag Cement"

_materials, 2020, doi:10.3390/ma13153265_

Round 1
Reviewer 1 Report
This reviewer would like to congratulate the research team on their efforts to research the effects of fly ash additions on the properties of concrete prepared using CEMIII cement. This is indeed a practice that needs better regulation in the European Union.
Table 2 shows properties of the cement used as declared by the producer but both cement types reach a compressive strength of approximately 50 Mpa which corresponds to CEMI cement please explain if this was considered when interpreting experimental results.
Figure 2 has icons instead of ions on the vertical axis title please correct, also on the horizontal axis values for CM1-33.2 are given while table 6 gives values for CM2-33.2 is this correct or should they be the same? Same for table 5.
Why are there no results presented for CM2, in the case of chloride ion migration, it would be interesting to see if there is a difference. If there are experimental results available, please present and discuss them.
Please review the English language and style of the paper there are many improvements that need to be made.
Reviewer 2 Report
Broad comments:
I found the article is well written; methodology and results are well described and graphically presented, there is a a sufficient number of references (35). So, I have minor demands on authors for improvements of this paper. I explain my suggestion and comments in more detail below:
The Introduction part is well written, with many references, but it is necessary to check/fullfil next:
- line 48, page 2:... instead of Rutkowska et al. (16) put ........Rutkowska and Małuszyńska (16)
- line 51, page 2:... instead of In paper (17) please add ........As reported Golewski (17)
- line 77, page 2:...please add name of standard in....Standard EN 206+A1:2016-12 (23) – same comment is for line 117, page 3.
- line 93, page 2:...author's name is Van Noort (26), please correct
- line 140, page 4:........In the German standard (28).. please add name of standard
Part 3. Test methods
- line 219, page 6:... determined in accordance with standard NT Build 492 (32) - please add; the same comment is for 227 line, page 6.
Reviewer 3 Report
The sustainability of concrete structures is a paramount issue and one of the most adopted strategies is by using additive materials, coming from the waste of other technological processes.
It is an emerging topic and of practical importance and interesting among the scientific community. The present research is quite well organized and well described. In this regard, it deserves the attention of the editorial board for considering its publications.
The results showed by the authors provide an insight into the benefits, from the physical and mechanical point of view, of adding FA in the cement or concrete compounds.
The percentage of the FA uses is not exhaustive in this research, and in further research maybe the authors could provide more data. However, the use of 25% and 33% is not fully justified by the authors.
The English require professional proofreading from what concerns the grammar.
Considering the above comments, I would suggest accepting the submitted manuscript.
